# Chatter Monitoring of Machining Center Using Head Stock Structural Vibration Analyzed with a 1D Convolutional Neural Network

**DOI:** 10.3390/s22145432

**Published:** 2022-07-20

**Authors:** Kwanghun Jeong, Yeonuk Seong, Jonghoon Jeon, Seongjun Moon, Junhong Park

**Affiliations:** 1School of Mechanical Engineering, Hanyang University, 222, Wangsimni-ro, Seongdong-gu, Seoul 04763, Korea; zjavbfk@hanyang.ac.kr (K.J.); tjddusdnr123@naver.com (Y.S.); rargon01@naver.com (J.J.); 2Machining Technology Research Group 2, Hwacheon Machine Tool Co., Ltd., Gwangju 62227, Korea; msj@hwacheon.com

**Keywords:** chatter detection, structural vibration, cepstral analysis, modal analysis, convolutional neural network

## Abstract

Real-time chatter detection is crucial for the milling process to maintain the workpiece surface quality and minimize the generation of defective parts. In this study, we propose a new methodology based on the measurement of machine head stock structural vibration. A short-pass lifter was applied to the cepstrum to effectively remove components resulting from spindle rotations and to extract structural vibration modal components of the machine. The vibration modal components include information about the wave propagation from the cutter impact to the head stock. The force excitation from the interactions between the cutter and workpiece induces structural vibrations of the head stock. The vibration magnitude for the rigid body modes was smaller in the chatter state compared to that in the stable state. The opposite variation was observed for the bending modes. The liftered spectrum was used to obtain this dependence of vibration on the cutting states. The one-dimensional convolutional neural network extracted the required features from the liftered spectrum for pattern recognition. The classified features allowed demarcation between the stable and chatter states. The chatter detection efficiency was demonstrated by application to the machining process using different cutting parameters. The classification performance of the proposed method was verified with comparison between different classifiers.

## 1. Introduction

Milling machines are widely used for the production of automobile and mobile components. The fully automated process with robot operations is cost-efficient without constraints from work hours. For effective production management, the metal cutting process must be monitored in real time. The physical responses of the machine tool are used for process diagnosis and for optimization of the cutting process. For automated optimization without a loss in production hours, chatter should be monitored and avoided by adjusting the cutting parameters, including the cutting depth, feed rate, and spindle rotation speed. To prevent chatter generation, the optimal cutting parameters should be determined during the process. When chatter occurs, it induces unstable vibration of the machine. The machine vibration amplitude fluctuates harmonically and generates relevant tonal sounds. Chatter vibration damages the machine parts by the strong transmission of the cutting force through the cutter and head stock.

Therefore, it is important to evaluate the cutting state through vibration measurement in real time and to control the cutting parameters to the optimal condition. Several studies have evaluated the structural parameters through experimental vibration measurements. The chatter generation was evaluated by measuring the acceleration, sound, and cutting bending moment responses. The bending moment response refers to the bending effect caused by a force at a point away from the pivot point [1]. Structural faults of a scale replica industrial plant were detected by edge devices in real time. The structural faults of the industrial plant were detected by edge devices using vibration and current sensors [2]. Mechanical and electrical monitoring was achieved using vibration and current sensors. The vibration response of a magnetorheological elastomer multifunctional grille composite sandwich plate was reduced through vibration measurement and active vibration control [3]. The frequency characteristics of the structure were examined to investigate the effect of active control. Real-time control of the parameters was successfully achieved using the vibrational response for structural integrity.

The frequency response function (FRF) from impact hammer tests was measured to determine the structural vibration characteristics of the cutter, cutter-holder, and spindle system [4]. The chatter feature was extracted based on the structural vibration characteristics. The vibration component originating from chatter generation was extracted by filter application to the resonance frequency bands of the system [5]. Time-frequency filtering was applied for chatter detection using statistical indicators [6]. The resonance frequency of the system was used to recognize the dominant chatter frequency bands [7]. FRF measurements were required for precise determination. To extract the chatter feature with minimal influence from the cutter, a modal analysis of the head stock was performed without the cutter.

During the machining process, the vibration response is dominated by harmonic components resulting from spindle rotation and the frictional impact between the cutter and workpiece. The vibration spectrum contains the excitation and transfer path components. A notch filter has been successfully used to remove the excitation components [8]. Since it is required to remove the excitation component independent of the spindle rotation speed, cepstral analysis was used. Cepstral analysis has the advantage of separating the excitation and transfer path components for feature extraction. The vibration components caused by harmonic excitations were removed to obtain contributions from the structural mode of a two-story framed structure [9]. The rubbing positions of the aero-engine were analyzed using the transfer path components [10]. The cepstrum transforms periodic harmonic components in the frequency domain into rahmonics in the quefrency domain. Short-pass liftering was used to remove the rahmonics from excitation and to extract the transfer path components in low quefrency [11]. A short-pass lifter was applied to the cepstrum, and the short-pass liftered cepstrum was calculated. A short-pass liftered spectrum was obtained by converting the short-pass-liftered cepstrum into a spectrum. The liftered spectrum was used to extract the modal characteristics of the head stock.

Machine learning classifiers of k-nearest neighbor (KNN), artificial neural networks (ANNs), and support vector machine (SVM) have been used for chatter detection [12,13]. The classifiers enable efficient chatter detection with small amounts of data and low computing power. Deep-learning-based classifiers were used for high-level feature extraction using a multi-layer structure [14]. Although the complexity due to the multi-layer network increases, large amounts of data and high computing power enable optimization of the algorithm. The convolutional neural network (CNN) shows excellent performance in image classification owing to its advantage in extracting features between adjacent data. The CNN classifier mimics the human visual system and consists of a convolutional layer, pooling layer, and fully connected layer [15]. The convolutional layer extracts relevant features. The pooling layer reduces and emphasizes the extracted features. In the convolutional layer, the input features are extracted using multiple filters with spatial information being maintained. Chatter detection was performed by a CNN classifier using two-dimensional data such as spectrograms, scalograms, and machined surface images in previous studies [16,17,18].

In this study, the vibration caused by force between the cutter and workpiece was used for chatter monitoring. The liftered spectrum successfully extracted the modal characteristics of the head stock regardless of the spindle rotation speed. Structural vibration of the head stock was analyzed to select features of the liftered spectrum. FRFs were measured at several points to analyze the mode frequency as well as the mode shape of the head stock. The mode shape was used to determine spectral characteristics of chatter vibration independent of the cutter and cutter-holder. Even when a different cutter was used, the features selected from the head stock structural properties were still effective. A 1D-CNN classifier was employed for cutting quality estimation. The 1D-CNN effectively extracted the characteristics of local information about the magnitude variations of resonance modes from the one-dimensional liftered spectrum. The 1D-CNN with the liftered spectrum was suitable for real-time chatter detection because it allowed much faster training and inference than typical CNNs that process two-dimensional data. It also enabled efficient memory management using only the features from the spindle head modes.

Figure 1 shows the scheme for real-time chatter detection at the machining center. Two industrial accelerometers were mounted on the head stock for vibration measurement in the *x*- and *y*-directions. A digital signal processor was used for data acquisition, signal processing, classification, and network communication. The vibration modal components were extracted using the cepstral analysis discussed in the following section. The cutting quality was monitored using the 1D-CNN classifier with the liftered spectrum. The classified result was transmitted to the FANUC controller of the computer numerical control machine by FOCAS communication.

## 2. Materials and Methods

### 2.1. Data Acquisition for the Milling Process under Stable and Chatter States

Experiments were performed during the milling process for the machining center. Figure 2 shows the experimental setup. Two industrial accelerometers (607A11 from PCB) were mounted on the head stock in the *x*- and *y*-directions, as shown in Figure 2a. The milling cutter had four flutes, a diameter of 10 mm, and an overall length of 47 mm, as shown in Figure 2b. Another milling cutter for verification of the proposed methodology had four flutes, a diameter of 16 mm, and an overall length of 67 mm, as shown in Figure 2c. The steel workpiece was machined as shown in Figure 2d. The machining was performed on (*A*) a flat shape in the diagonal direction, (*B*) a flat shape in the *x*-direction, (*C*) a curved shape in the *y*-direction, and (*D*) a flat shape in the *y*-direction. This allowed data acquisition for the cutter and workpiece interaction in various directions. The vibration response was processed using a Raspberry Pi 4 model B with a sampling frequency of 44,100 Hz, as shown in Figure 2e. The cutting parameters and states in the milling tests are listed in Table 1. The spindle rotation speed was varied from 2000 to 6200 rpm. The radial depth was varied between 0.1 and 0.2 mm, with an axial depth of 12 mm and feed rate of 1500 mm/min. The data from each experiment were labeled with the stable or chatter states. The vibration characteristics depended on the cutting parameters. The spindle rotation speed had a particularly large influence on the cutter vibrations. Because deep learning has excellent performance on classification, vibration responses under various spindle rotation speeds were required for robust classification. Figure 3 shows the machined surface of the workpiece under stable and chatter states. The surface image consisted of a vertical and diagonal waveform on the stable and chatter machined surface, respectively. In each experiment, the specific state was maintained during cutting. The state of each experiment was labeled based on the surface cutting patterns.

### 2.2. Cepstral Analysis for Feature Extraction

The complex cepstrum is obtained as the inverse Fourier transform of the log spectrum of the vibration response in the time domain, as follows:(1)$$C_{c}\left(\tau \right)=F^{-1}\left[\ln\left(X\left(\omega \right)\right)\right],$$
where τ is the quefrency and X is the spectrum calculated as
(2)X(ω)=F[x(t)]=A(ω)exp(jϕ),
where *A* and *ϕ* are the amplitude and phase of the vibration response, and *F* denotes the Fourier transform.

The real cepstrum is obtained by setting the phase to zero in Equation (2), as follows:(3)$$C_{r}\left(\tau \right)=F^{-1}\left[\ln\left(A\left(\omega \right)\right)\right].$$

The vibration response acquired in the mechanical system is the convolution between the excitation e(t) and transfer path response h(t) of the mechanical system, expressed as follows:(4)x(t)=e(t)∗h(t),
where ∗ is the convolution operation.

With the contribution from the transfer paths, the cepstrum is obtained as
(5)$$C_{r}\left(\tau \right)=F^{-1}\left[\ln\left(\left|X\left(\omega \right)\right|\right)\right]=F^{-1}\left[\ln\left(\left|E\left(\omega \right)H\left(\omega \right)\right|\right)\right]=F^{-1}\left[\ln\left(\left|E\left(\omega \right)\right|\right)\right]+F^{-1}\left[\ln\left(\left|H\left(\omega \right)\right|\right)\right].$$

The obtained cepstrum was used to separate the excitation and transfer path components. The transfer path component was concentrated at a low quefrency. The excitation component appeared at a relatively high quefrency. Short-pass liftering was applied to the real cepstrum to extract the transfer path component concentrated at a low quefrency. The spindle rotation components were eliminated, and the vibration modal components were obtained. The short-pass liftered cepstrum $C_{l}\left(\tau \right)$ was calculated by multiplying the real cepstrum as follows:(6)$$C_{l}\left(\tau \right)=C_{r}\left(\tau \right)L_{s}\left(\tau -\tau_{c}\right),$$
where $\tau_{c}$ is the cut-off quefrency and $L_{s}$ is the short-pass lifter ($L_{s}=1$ when $\tau \leq \tau_{c}$, and =0 elsewhere).

The short-pass-liftered cepstrum and the liftered spectrum were calculated as shown in Figure 4. The liftered spectrum was obtained by applying the Fourier transform and exponential function to the short-pass liftered cepstrum. The liftered spectrum was used to analyze the vibration responses under stable and chatter states.

### 2.3. Classification by 1D Convolutional Neural Network

A CNN is a network structure stacked in various layers, including linear and nonlinear operations. In this study, a 1D-CNN classifier was employed to extract local information from one-dimensional data. The classifier consisted of an input layer, three convolution layers, a fully connected layer, and an output layer, as shown in Figure 5.

Data normalization prevents the classifier domination by specific feature values. Min–max normalization was used to convert all feature values in the input data to a range between 0 and 1. Normalized data were fed into the nodes of the input layer. The output of the input layer was used as the input for the subsequent convolutional layer. The layer performed a convolution on the input data using a filter bank consisting of several learnable filters. If a filter bank with n filters was used as the weight in the convolution layer, the output of the convolution layer had n channels. The convolution operation was used to extract the salient local spatial features of the input data. The weights in the filter bank were applied to all regions of the input data for weight sharing. The weight sharing reduced the complexity of the network due to fewer learning parameters than ANNs.

The k-th filter output $x_{k}{}^{l}$ of convolutional layer l was calculated as follows:(7)$$x_{k}{}^{l}=\sum_{c=1}^{n}w_{c,\;k}{}^{l}*s_{c}{}^{l}+b_{k}{}^{l},$$
where $s_{c}{}^{l}$ denotes the input of channel c of layer l, $w_{c,k}{}^{l}$ denotes the weight of channel c of filter k in layer l, n denotes the number of input channels, $b_{k}{}^{l}$ denotes the bias vector of filter k in layer l, and the notation ∗ rep resents the convolution operator.

The rectified linear units (ReLU) activation function improved the data representation ability. The activation function was applied to the output after convolution. The ReLU applied output was the final outcome of the convolutional layer. The ReLU activation was performed as follows:(8)f(z)=max(0,z).

The last convolutional layer output was flattened and used as the input for the fully connected layer. The fully connected layer correlates all nodes in the previous layer to those in the subsequent layer. The nodes were independent of each other without any connection. The j-th node output $x_{j}{}^{l}$ in layer l was calculated as follows:(9)$$y_{k}=\sum_{c=1}^{m}w_{c,\;k}*x_{c}+b_{k}\;\;\;\;\;\;k=1,2,\ldots n\;,$$
where $s_{i}{}^{l}$ denotes the i-th node input of layer l, $w_{ij}{}^{l}$ denotes the scalar weight between the i-th node in layer l and j-th node in layer l, and $b_{j}{}^{l}$ denotes the bias scalar of the i-th node in layer l.

The output layer structure was the same as that of the fully connected layer. The output layer represented the classification result. The predicted probability distribution from the output layer was compared to the actual probability distribution. The softmax activation function was applied to the output layer values. The softmax is expressed as
(10)$$q\left(z_{j}\right)=e^{z_{j}}/\sum_{k=1}^{n_{e}}e^{z_{k}},$$
where $z_{j}$ denotes the j-th element value in vector z and $n_{e}$ is the number of elements.

We used cross-entropy loss as the loss function and the backpropagation algorithm for training. Backpropagation calculates the gradient of loss and adjusts all weights such that the predicted probability distribution is equal to the actual probability distribution [19].

## 3. Results

### 3.1. Modal Properties of the Machine Tool Head Stock

The machine tool head stock was derived in the *z*-direction during the metal cutting. The head stock structure was designed to be as rigid as possible. Despite the large stiffness in the longitudinal direction, its structural vibration was not completely prevented. The cutter and workpiece interaction induced structural vibration. Undesirable flexural vibration of the head stock induced the chatter cutting state. To investigate the vibration generation during the metal cutting process, the modal properties were measured using accelerometers (352A-21, PCB) with the vibration excitation of an impact hammer (5802AT, Dytran). Figure 6 shows the measured vibration FRF at the cutter tip. The vibration resonance occurred at 3000 Hz. The cantilever beam mode contributed substantially to the cutter vibration response. To investigate the structural mode, a modal test was performed on the head stock. The accelerometers were attached as shown in Figure 7. The impact FRF was obtained in the frequency range of 0–6400 Hz. The modal properties in both the *x*- and *y*-directions were obtained. Figure 8 shows the measured vibration FRFs of the head stock in the *x*- and *y*-directions. Vibration resonances occurred at 770, 3840, 4280, 4900, and 5250 Hz in the *x*-direction and at 150, 770, 3840, 4280, 5070, and 5750 Hz in the *y*-direction. Resonance frequencies of 770, 3840, and 4280 Hz were present in both directions. Figure 9 shows the mode shapes of the head stock. The rigid body mode occurred at 770 Hz and the bending mode occurred at 3840 and 4280 Hz in both directions. In the *x*-direction, the bending mode occurred at 4900 and 5250 Hz. In the *y*-direction, the rigid body mode occurred at 150 Hz and the bending mode occurred at 5070 and 5750 Hz.

### 3.2. Feature Extraction by Cepstral Analysis

During the metal cutting, the machine vibration response was influenced by the modal response of the head stock. The modal components were extracted from the measured acceleration using the cepstral analysis as shown in Figure 10. The vibration was measured at the spindle rotation speed of 3200 rpm. Figure 10a shows the Hanning-windowed vibration response. Figure 10b,c show its log spectrum and cepstrum, respectively. The spindle rotation components appeared as tones in the log spectrum. They were converted into tones and rahmonics in the cepstrum. The fundamental rahmonic appeared at 0.0188 s (53.33 Hz) in the quefrency domain. Its rahmonics appeared at multiples of the fundamental rahmonic quefrency. Figure 10d,e show the short-pass-liftered cepstrum and liftered log spectrum, respectively. A short-pass lifter with a cut-off value of 0.00599 s was designed. The lifter was applied to remove spindle rotation components below 10,000 rpm. The rahmonics were removed using a lifter in the liftered cepstrum. The vibration modal components contributed to the non-periodic region of the log spectrum and the liftered log spectrum. Figure 10f shows the original spectrum and the liftered spectrum. The liftered spectrum contained vibration modal components without contribution from the spindle rotation and cutter.

This approach was applied to the measured vibration response for both the stable and chatter states (stable state: spindle rotation speed was 3200 rpm, radial depth was 0.2 mm, axial depth was 12 mm, and feed rate was 1500 mm/min; chatter state: spindle rotation speed was 5000 rpm, radial depth was 0.1 mm, axial depth was 12 mm, and feed rate was 1500 mm/min). Figure 11 and Figure 12 show the vibration response in the *x*-direction and its liftered spectrum with cutting sections *A*–*D* in Figure 4. For the stable state, the liftered spectrum levels in sections *A*–*D* were 0.51, 0.46, 0.60, and 0.58 m/s^2^, respectively. For the chatter state, the levels were 0.63, 0.57, 0.65, and 0.67 m/s^2^, respectively. In sections *A* and *B*, similar vibration responses appeared because the machining system vibrated strongly in the *y*-direction. Conversely, in sections *C* and *D*, similar vibration responses appeared because the machining system vibrated significantly in the *x*-direction. Therefore, the vibration response measured in the *x*-direction was larger in sections *C* and *D*. This increased vibration level from the stable state to the chatter state was mostly contributed from the vibration modes of the head stock in the *x*-direction.

The cutter vibration mode appeared at 3000 Hz. In this frequency, there was little difference in vibration magnitude between the stable and chatter states. As the accelerometers were mounted on the head stock, the influence of the head stock vibration modes was relatively greater than that of the cutter vibration modes. Compared with the bending modes in Figure 9a, the vibration magnitude in the chatter state was larger than that in the stable state at 3840 and 4280 Hz. The effect of the bending modes also affected the vibration at the frequency range from 3600 to 4400 Hz. The bending modes appeared both in the *x*- and *y*-directions. The magnitude in the corresponding frequency range was larger in sections *C* and *D* than in sections *A* and *B*. These vibration characteristics were contributed by the vibration measured in the *x*-direction.

Compared with the mode in the *x*-direction (Figure 9b), the vibration magnitude in the chatter state was smaller than that of the stable state at 770 Hz and larger than that of the stable state at 4900 and 5250 Hz. For the rigid body mode at 770 Hz, the magnitude measured in the chatter state was smaller. For the bending modes at 4900 and 5250 Hz, the vibration magnitude in the chatter state was larger. These head stock modes in the *x*-direction exhibited similar variation regardless of the cutting directions.

As shown in Figure 9c, the head stock mode in the *y*-direction was coupled to the those in the *x*-direction. The coupling magnitude was noticeable in the bending mode at 5750 Hz. In sections *A* and *B* (the head stock vibrations were larger in the *y*-direction), the effect of the bending mode was greater than that in sections *C* and *D* (the head stock vibrated significantly in the *x*-direction). In all sections, the vibration magnitude in the chatter state was larger than that in the stable state at the frequencies of the bending modes. This characteristic was especially significant for the process in sections *A* and *B*. When chatter vibration was generated, the kinetic energy transferred from the rigid body mode in the low-frequency region to the bending mode in the high-frequency region. As the cutting state changed from stable to chatter, the vibration magnitude decreased in the rigid body modes and increased in the bending modes.

Figure 13 and Figure 14 show the vibration response measured in the *y*-direction and its liftered spectrum, respectively. The vibration characteristics were similar to those measured in the *x*-direction. For the stable state, the liftered spectrum levels were 0.58, 0.57, 0.52, and 0.48 m/s^2^, respectively in sections *A*, *B*, *C*, and *D*. For the chatter state, the levels were 0.63, 0.64, 0.60, and 0.61 m/s^2^, respectively. The vibration response measured in the *y*-direction was larger in sections *A* and *B* where the machining system vibrated in the *y*-direction. This increased vibration level during transition from the stable to the chatter states was contributed from the vibration modes of the head stock in the *y*-direction. When compared with the mode frequency in Figure 9a,c, the magnitude at the chatter state was smaller than that at the stable state for rigid body modes in 150 and 770 Hz. The vibration response was larger than that in the stable state at 3840, 4280, 5070, and 5750 Hz, where the bending mode contributed to the vibration response.

The natural frequencies of the head stock were identified at the frequency of 100–800 Hz for the rigid body modes and 3600–6400 Hz for the bending modes. The original spectrum and liftered spectrum of the mode frequency were visualized using t-stochastic neighbor embedding (t-SNE) to identify the effectiveness of the input features in chatter detection, as shown in Figure 15. The t-SNE visualized high-dimensional data by reducing it to two-dimensional data [20]. The original spectral data set was not clustered into stable and chatter states, but clustered according to the cutting parameters. The clustering result using the liftered spectral data set showed better performance in the detection of the cutting state than the one by the original spectral data set. Therefore, the liftered spectrum was selected as the input for the chatter detection.

### 3.3. Feature Generation from the Liftered Spectrum

For data-based chatter classification, the vibration response must be converted into a liftered spectrum for different operation parameters. Each experiment consisted of 187 liftered spectra, and the total number of liftered spectra was 2431, as shown in Table 1. The acquired data were randomly divided into training, validation, and test sets for the classifier model. To obtain the reliability of the chatter detection for machining operations, the data were divided such that the parameters varied between the training and test sets. Three machining experiments in the stable state and two experiments in the chatter state were used as the test set. The liftered spectrum and label from the remaining eight experiments were randomly divided into training and validation sets at a 7:3 ratio. A liftered spectrum was selected as the input to the classifier. Because the frequency resolution was 5.38 Hz, the input data size was 650.

### 3.4. Chatter Detection Using the Proposed Procedure

The 1D-CNN classifier shown in Table 2 was employed. The filter size of the convolutional layers was three. The number of filters in the first, second, and third convolution layers was 10, 20, and 30, respectively. The ReLU activation function was used after every convolutional layer and the fully connected layer. The flatten layer was used to change the shape of the multidimensional matrix for operations with nodes in the fully connected layer. A dropout layer with a 50% rate was used after the fully connected layer to prevent network overfitting. The softmax activation function was used in the output layer for classification. The Adam optimizer with a learning rate of 0.0001 and mini-batch size of 1000 was used. The proposed classifier was trained for 30 epochs using the liftered spectrum for chatter detection. Figure 16 shows the performance of the classifier on the training and validation sets. During training, the classification accuracy increased with decreasing cross-entropy loss. The training and validation accuracies were 90.7% and 91.3% at 15 epochs. The training and validation accuracies reached 99.5% and 100.0% at 30 epochs. There was no overfitting, as shown by the cross-entropy loss between the training and validation sets.

Table 3 lists the confusion matrix of the classifier on the test set. The classifier had a recall of 100.0%, a precision of 97.4%, an F-score of 98.7%, and an accuracy of 98.4%. The classifier had good performance under the cutting parameters not used for training. The cutting process was classified as a stable or chatter state using the output of the classifier. Figure 17 shows the probability density function of the output of the chatter node on the test set. When the output exceeded 0.5, the machining process was diagnosed as the chatter state. The output in the stable state was concentrated at 0.2–0.3 and that in the chatter state was widely distributed at 0.5–0.8. Considering that the output in the stable state is less than 0.4, it is possible to diagnose chatter generation when the output is larger than 0.4. Figure 18 shows t-SNE using the fully connected layer output of the 1D-CNN classifier. The output set was clearly clustered depending on the cutting states. This indicated that the proposed procedure is efficient for chatter detection regardless of the cutter used in the machining operation.

The classification performance of the proposed 1D-CNN process was compared to that of other methodologies, including KNN, ANN, SVM, and DNN [21,22]. The original spectrum was used as a useful feature for fault detection. The classification results using the original spectrum were also analyzed for comparison with the liftered spectrum. The accuracy using the original spectrum and liftered spectrum are compared in Table 4. The KNN classifier adopted the Euclidean distance as the similarity function and the number of nearest neighbors was three. The ANN classifier adopted a hidden layer of 700 nodes and ReLU activation function. The linear SVM (L-SVM) and RBF-SVM classifiers used a linear function and a radial basis function (RBF) as kernels, respectively [23]. The DNN classifier adopted three hidden layers with 800, 500, and 200 nodes, respectively. The ReLU activation function was applied to the outputs of all hidden layers. The 1D-CNN classifier had the same architecture as the proposed classifier. The liftered spectrum is an outstanding feature for chatter detection. The 1D-CNN classifier with a liftered spectrum exhibited better performance than the other classifiers.

Application and verification of the proposed methodology in actual metal cutting was performed using the Raspberry Pi 4 model B in the Appendix A. It took about 0.2 s to acquire the vibration response of 8192 samples, 0.1 s to calculate the liftered spectrum, and 0.1 s for classification using the 1D-CNN. For the stability of the system, the state of the vibration response was checked using the classification every 0.5 s. The spindle rotation speed was maintained when the vibration response was classified as the stable state. The spindle rotation speed changed when the machining was classified as the chatter state. The computational cost of 0.1 s for the proposed 1D-CNN made the near real-time chatter detection possible.

### 3.5. Classification for Different Structural Vibration Characteristics—Machining by a Different Cutter

When a different cutter is used, the vibration characteristics change depending on the tool mechanical properties. The liftered spectrum shows the vibrational modal components of the system. The feature extraction was performed using the identified modal vibration responses. Since the dependence of the machine structural vibration was used for classification, the effects of the cutter were not significant. This is advantageous to construct a deep learning classifier with a relatively small amount of data. To validate the advantage, the proposed procedure was applied to the metal processing using the different cutter, as shown in Figure 2c. Figure 19 shows the liftered spectrum of the vibration response at the cutting section *A* for the stable and chatter states (stable state: spindle rotation speed was 3000 rpm, radial depth was 0.1 mm, axial depth was 20 mm, and feed rate was 1500 mm/min; chatter state: spindle rotation speed was 3600 rpm, radial depth was 0.1 mm, axial depth was 20 mm, and feed rate was 1500 mm/min). Although the different cutter was used, the head stock natural frequency remained the same. For the rigid body mode, the vibration magnitude in the chatter state was smaller than that in the stable state. In the bending modes, the opposite variation was observed.

The cutting parameters and states in the milling tests using the different milling cutter are listed in Table 5. The spindle rotation speed was varied from 1500 to 4800 rpm. The radial depth was 0.1 mm, with an axial depth of 15 mm and feed rate of 1500 mm/min. Three machining experiments in the stable state and two experiments in the chatter state were used as the test set. The liftered spectrum and label from the remaining seven experiments were randomly divided into training and validation sets at a 7:3 ratio. The 1D-CNN classifier was trained for 30 epochs using the liftered spectrum for chatter detection. Figure 20 shows the performance of the classifier on the training and validation sets. During training, the classification accuracy increased with decreasing cross-entropy loss. The training and validation accuracies reached 96.9% and 98.2% at 30 epochs. Table 6 lists the confusion matrix of the classifier on the test set with the different milling cutter. The classifier had a recall of 99.6%, a precision of 94.6%, an F-score of 97.0%, and an accuracy of 96.4%. The methodology was successfully applied to detect chatter in the mechanical system using the different milling cutter. This result indicated that chatter detection for various milling cutters was possible through the measurement of the vibration characteristics of the head stock.

## 4. Conclusions

For metal cutting, an experienced operator notices the change in cutting states via the vibration generated by the machine. In this study, a new methodology for real-time chatter detection was proposed based on vibration measurements on the machine tool head stock. The measured vibration was processed via cepstral analysis and a 1D-CNN to evaluate the machining quality. Short-pass liftering was used to extract the modal vibration response of the mechanical system. The liftered spectrum contained the vibration properties of the milling cutter and spindle structure with minimal influence from the spindle rotation components. An experimental modal analysis was performed to determine the structural vibration characteristics. The vibration response of the head stock was influenced by the structural vibration characteristics. For the rigid body mode, the vibration magnitude was reduced with the generation of chatter. For the flexural bending modes, the modal vibration magnitude increased with the chatter generation. With the chatter generation, the vibration energy was transferred from the rigid body mode in the low-frequency region to the bending mode in the high-frequency range. The chatter generation was identified using the observed spectral characteristics. The liftered spectrum of the mode frequency was used as the input feature for the chatter detection. The proposed method based on the 1D-CNN performed better than the other classifiers. The reduction in the rotation harmonic components significantly increased the reliability of the classified results. The chatter detection was successfully achieved using various cutting parameters and cutting sections. Furthermore, the proposed methodology was successfully applied to detect chatter for machining process using a different milling cutter. Since a wide variety of tools are used for material removal using a machining center, the proposed classifier is advantageous in minimizing the influence from the cutter for cutting state identification. The methodology significantly reduces the data required for classification and is advantageous for in-process monitoring of machining quality.

## Figures and Tables

**Figure 1 sensors-22-05432-f001:**
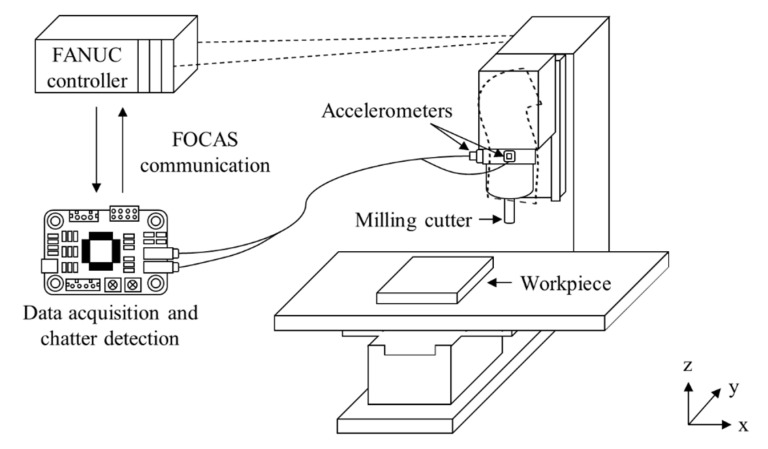
Scheme for online chatter detection at the machining center.

**Figure 2 sensors-22-05432-f002:**
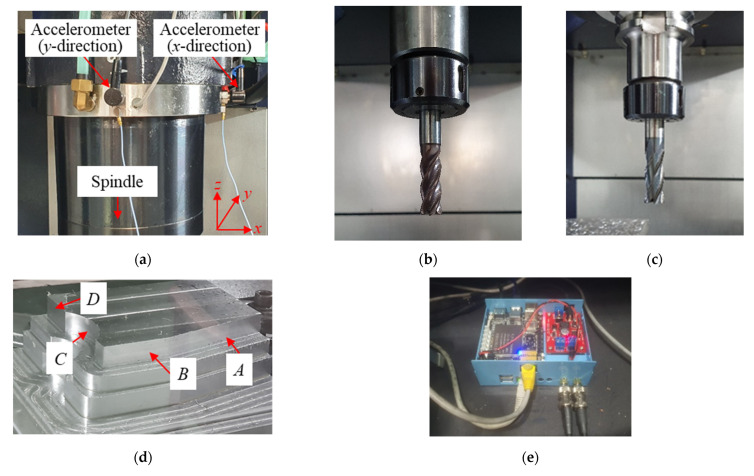
Experimental setup: (**a**) accelerometers mounted on the head stock in the *x*- and *y*-directions, (**b**) milling cutter, (**c**) another milling cutter for verification of the proposed methodology, (**d**) steel workpiece with four different sections (*A*–*D*), and (**e**) integrated board for data acquisition, signal processing, classification, and communication.

**Figure 3 sensors-22-05432-f003:**
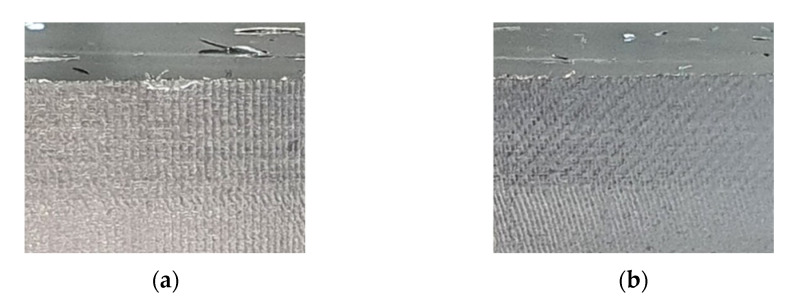
Surface of the machined workpiece under (**a**) stable and (**b**) chatter states.

**Figure 4 sensors-22-05432-f004:**
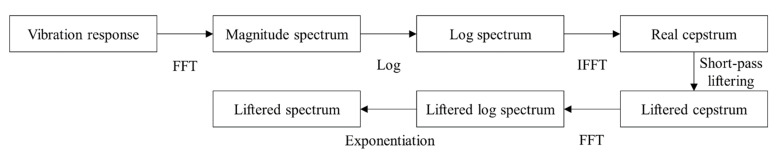
Calculation procedure for short-pass-liftered cepstrum and liftered spectrum.

**Figure 5 sensors-22-05432-f005:**
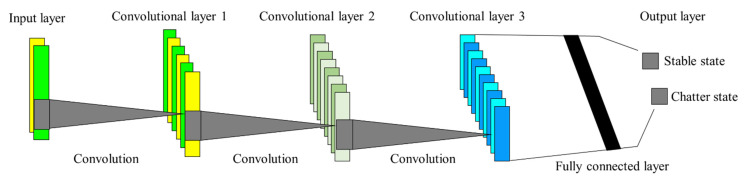
One-dimensional CNN architecture used to classify stable and chatter states. The 1D-CNN consists of an input layer, three convolution layers, a fully connected layer, and an output layer.

**Figure 6 sensors-22-05432-f006:**
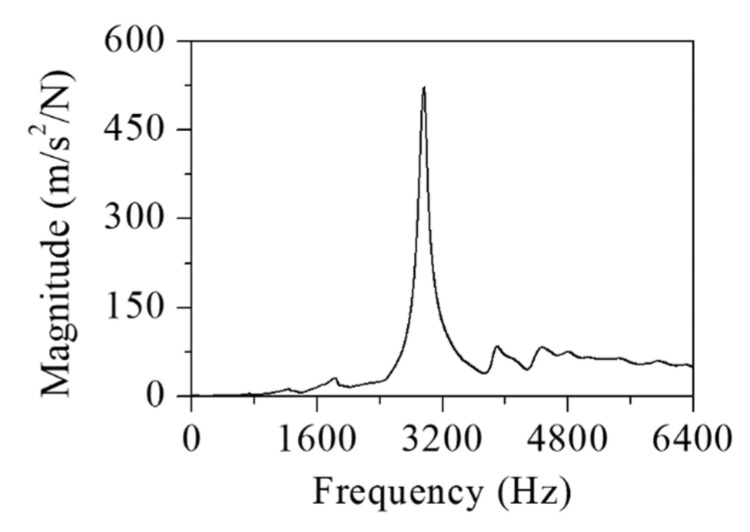
Measured vibration FRF at the cutter tip.

**Figure 7 sensors-22-05432-f007:**
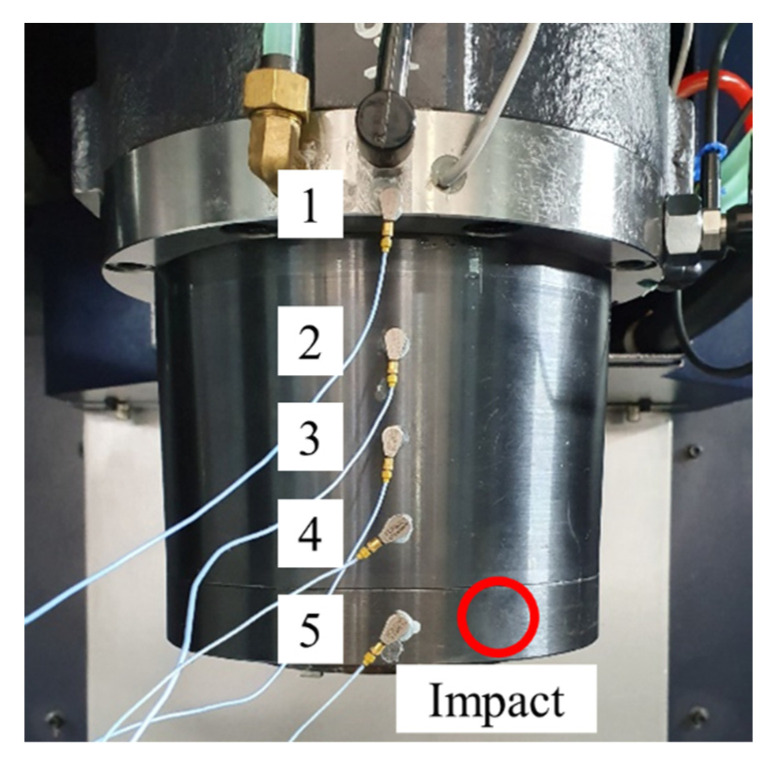
Experimental setup for modal test of the head stock. Accelerometers were attached at five locations and the number indicated the attachment location.

**Figure 8 sensors-22-05432-f008:**
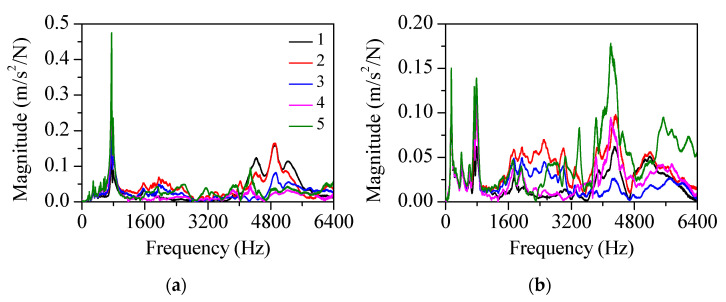
FRFs of the head stock at five locations in the (**a**) *x*- and (**b**) *y*-directions.

**Figure 9 sensors-22-05432-f009:**
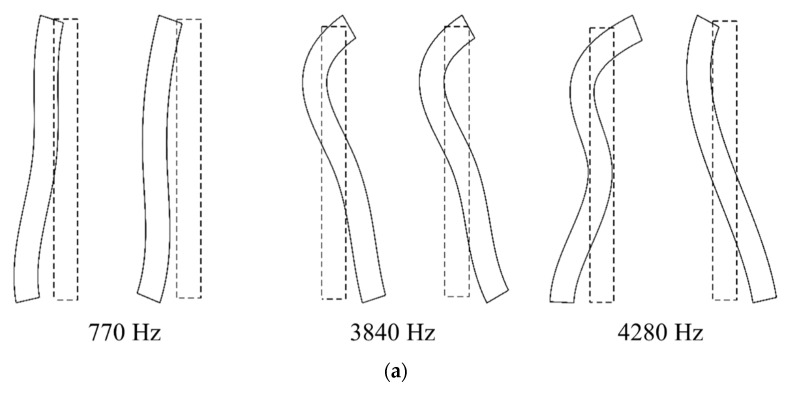
Mode shapes of the head stock in the (**a**) *x*- and *y*-directions (*x*: left, *y*: right), (**b**) *x*-direction, and (**c**) *y*-direction.

**Figure 10 sensors-22-05432-f010:**
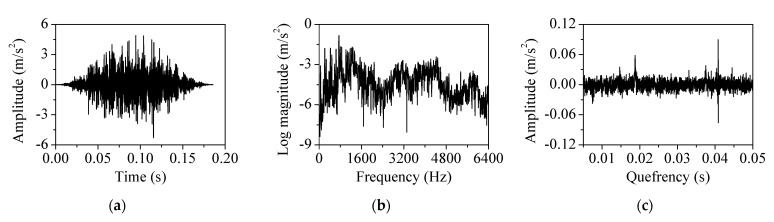
Cepstral analysis using short−pass liftering for feature extraction: (**a**) Hanning−windowed vibration response, (**b**) log magnitude spectrum, (**c**) cepstrum, (**d**) short−pass liftered cepstrum, (**e**) liftered log magnitude spectrum, and (**f**) original spectrum (black) and liftered spectrum (red).

**Figure 11 sensors-22-05432-f011:**
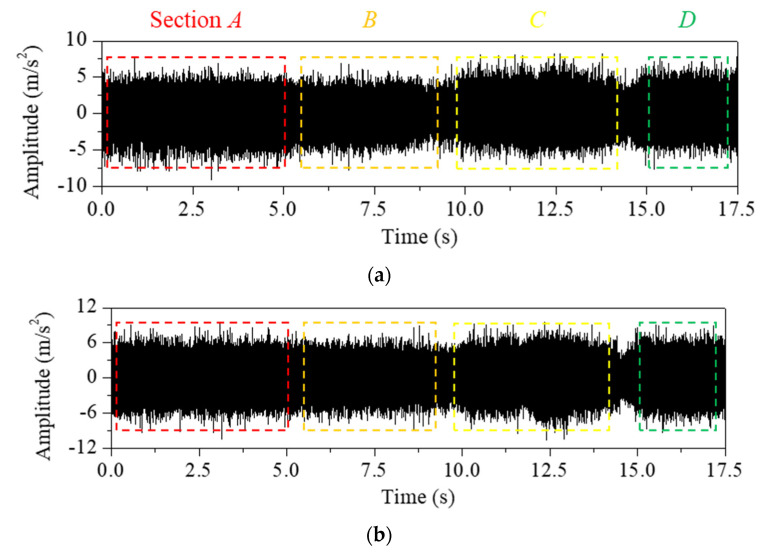
Measured vibration response in the *x*−direction at four cutting sections *A*, *B*, *C*, and *D*. (**a**) Stable state: spindle rotation speed was 3200 rpm, radial depth was 0.2 mm, axial depth was 12 mm, and feed rate was 1500 mm/min. (**b**) Chatter state: spindle rotation speed was 5000 rpm, radial depth was 0.1 mm, axial depth was 12 mm, and feed rate was 1500 mm/min.

**Figure 12 sensors-22-05432-f012:**
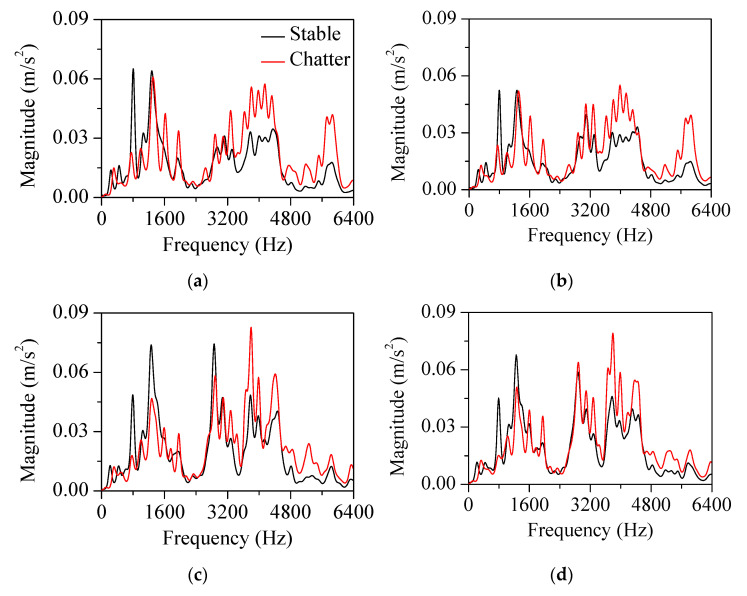
Liftered spectrum of the vibration response measured in the *x*-direction at cutting sections: (**a**) section *A*, (**b**) *B*, (**c**) *C*, and (**d**) *D*.

**Figure 13 sensors-22-05432-f013:**
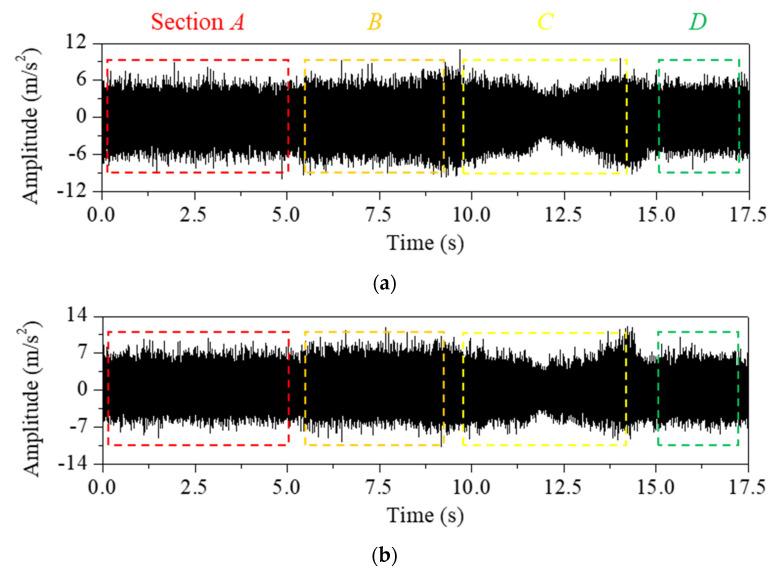
Measured vibration response in the *y*−direction at four cutting sections *A*, *B*, *C*, and *D*. (**a**) Stable state: spindle rotation speed was 3200 rpm, radial depth was 0.2 mm, axial depth was 12 mm, and feed rate was 1500 mm/min. (**b**) Chatter state: spindle rotation speed was 5000 rpm, radial depth was 0.1 mm, axial depth was 12 mm, and feed rate was 1500 mm/min.

**Figure 14 sensors-22-05432-f014:**
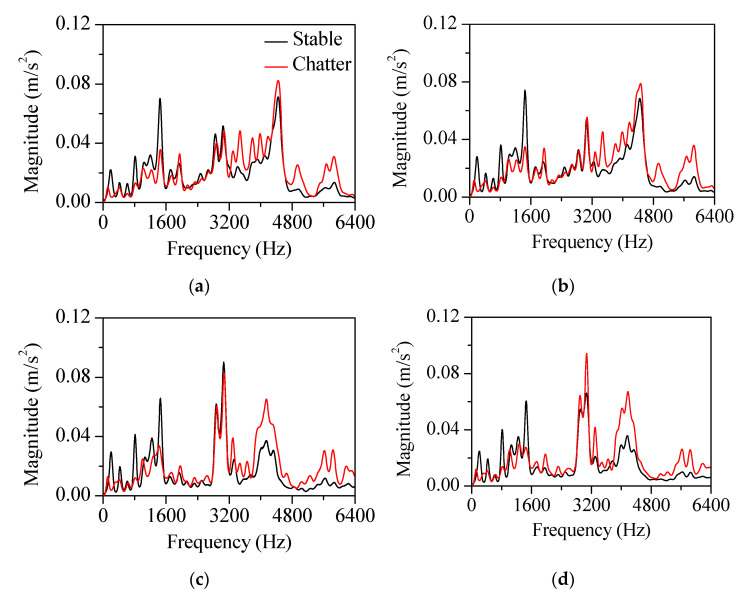
Liftered spectrum of the vibration response in the *y*−direction at cutting sections: (**a**) section *A*, (**b**) *B*, (**c**) *C*, and (**d**) *D*.

**Figure 15 sensors-22-05432-f015:**
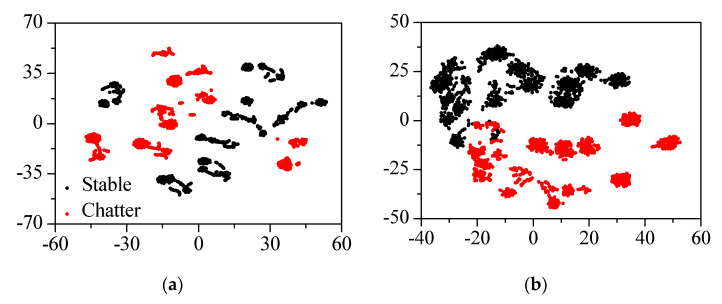
The t−SNE of (**a**) original spectrum and (**b**) liftered spectrum of the vibration response.

**Figure 16 sensors-22-05432-f016:**
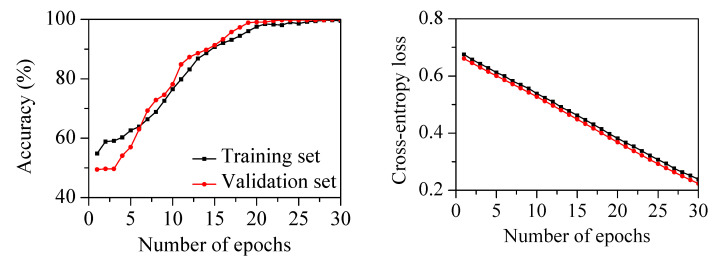
Accuracy and cross-entropy loss of the classifier on training and validation sets with increasing number of epochs.

**Figure 17 sensors-22-05432-f017:**
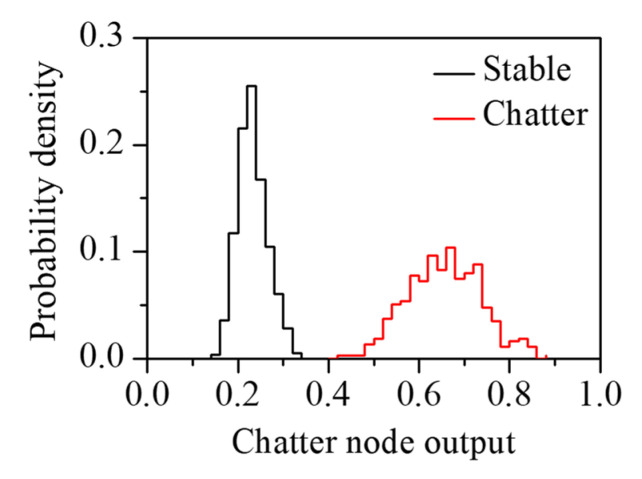
Probability density function of the output of chatter node on the test set.

**Figure 18 sensors-22-05432-f018:**
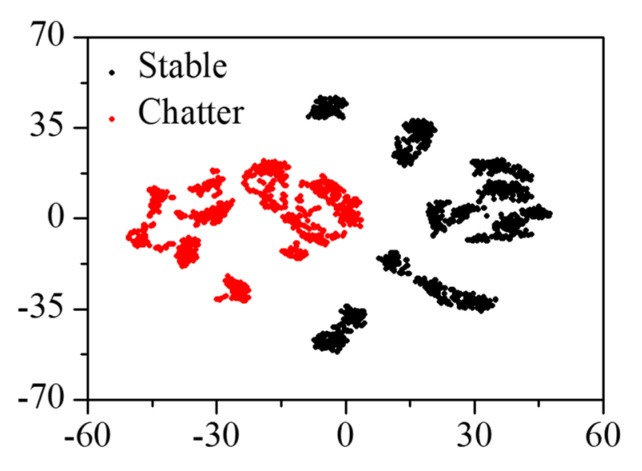
The t−SNE using fully connected layer output of the 1D−CNN classifier.

**Figure 19 sensors-22-05432-f019:**
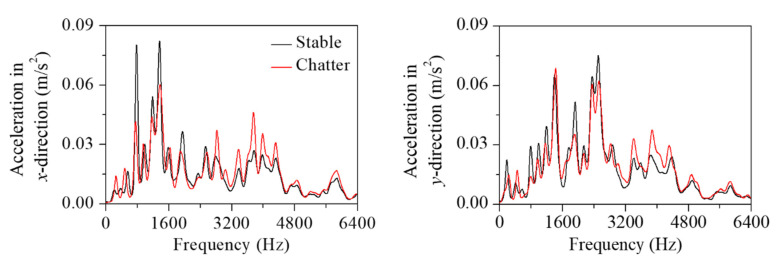
Liftered spectrum of the vibration response at cutting section *A* using the different milling cutter.

**Figure 20 sensors-22-05432-f020:**
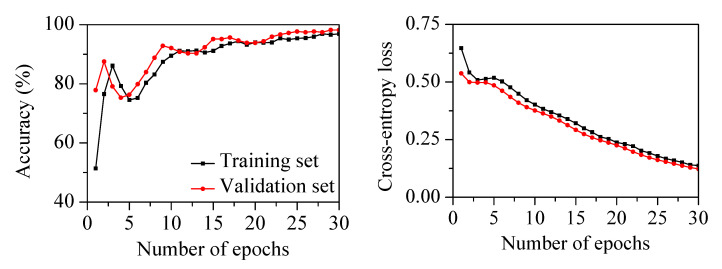
Accuracy and cross-entropy loss of the classifier on training and validation sets with increasing number of epochs using the different milling cutter.

**Table 1 sensors-22-05432-t001:** Cutting parameters and states in the milling tests.

Speed (rev/min)	Radial Depth (mm)	Axial Depth (mm)	Feed Rate (mm/min)	State
2000, 2300, 2600	0.1	12	1500	Stable
2900, 3200, 3500, 4100	0.2	12	1500	Stable
4700, 5000, 5300	0.1	12	1500	Chatter
5600, 5900, 6200	0.2	12	1500	Chatter

**Table 2 sensors-22-05432-t002:** Detailed architecture of the proposed classifier.

Layer	Input Shape	Output Shape
Input layer	650, 2	650, 2
Convolutional layer (ReLU)	650, 2	324, 10
Convolutional layer (ReLU)	324, 16	161, 20
Convolutional layer (ReLU)	161, 32	80, 30
Flatten layer	80, 30	2400
Fully connected layer (ReLU)	2400	256
Dropout layer	256	256
Output layer (softmax)	256	2

**Table 3 sensors-22-05432-t003:** Confusion matrix of the test set.

		Predicted State	
		Stable	Chatter
Actual state	Stable	561	0
	Chatter	15	359

**Table 4 sensors-22-05432-t004:** Prediction accuracy (%) using the original spectrum and liftered spectrum.

		Classifier					
		KNN	ANN	L-SVM	RBF-SVM	DNN	1D-CNN
Input	Original spectrum	68.78	63.4	79.8	76.2	49.7	73.7
	Liftered spectrum	81.4	90.5	82.4	91.8	87.8	98.4

**Table 5 sensors-22-05432-t005:** Cutting parameters and states in the milling tests using the different milling cutter.

Speed (rev/min)	Radial Depth (mm)	Axial Depth (mm)	Feed Rate (mm/min)	State
1500, 1800, 2100, 2400, 2700, 3000, 3300	0.1	15	1500	Stable
3600, 3900, 4200, 4500, 4800	0.1	15	1500	Chatter

**Table 6 sensors-22-05432-t006:** Confusion matrix of the test set with the different milling cutter.

		Predicted State	
		Stable	Chatter
Actual state	Stable	559	2
	Chatter	32	342

## Data Availability

Not applicable.

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
