# Peer review of "Chatter Monitoring of Machining Center Using Head Stock Structural Vibration Analyzed with a 1D Convolutional Neural Network"

_sensors, 2022, doi:10.3390/s22145432_

Round 1

Reviewer 1 Report

The paper is interesting but there are some issues that must be improved. The main is related to the innovation content of the research. Please motivate how the proposed approach is different from the ones already used based on convolutional neural networks. For spindle speed a wide range of values was used. Please provide a motivation for this choise.

In the conclusions, the authors affirm that the proposed approach allows a real-time chatter detection using a raspberry PI4 board . Please provide computational costs and timing diagrams, also respect to other existing algorithms, to asses the effective real-time applicability of proposed method

Other minor issues are:

Raw 14: "cestrum" should be cepstrum

Raw 207: remove space "rep resents"

Round 2

Reviewer 1 Report

C1

row 520: with 0.1s of computational the use  "Real-time" adjective it is not correct, please replace "real-time" with "near real-time" as you  introduce a time delay for feedback and control purposes.

Author Response

We replaced "real-time" with "near real-time".
